# Application of Proteogenomics to Urine Analysis towards the Identification of Novel Biomarkers of Prostate Cancer: An Exploratory Study

**DOI:** 10.3390/cancers14082001

**Published:** 2022-04-15

**Authors:** Tânia Lima, António S. Barros, Fábio Trindade, Rita Ferreira, Adelino Leite-Moreira, Daniela Barros-Silva, Carmen Jerónimo, Luís Araújo, Rui Henrique, Rui Vitorino, Margarida Fardilha

**Affiliations:** 1Department of Medical Sciences, Institute of Biomedicine-iBiMED, University of Aveiro, 3810-193 Aveiro, Portugal; tanialima@ua.pt (T.L.); rvitorino@ua.pt (R.V.); 2Cancer Biology and Epigenetics Group, Research Center of Portuguese Oncology Institute of Porto (GEBC CI-IPOP) & Porto Comprehensive Cancer Center (P.CCC), 4200-072 Porto, Portugal; daniela.silva@ipoporto.min-saude.pt (D.B.-S.); carmenjeronimo@ipoporto.min-saude.pt (C.J.); henrique@ipoporto.min-saude.pt (R.H.); 3UnIC@RISE, Department of Surgery and Physiology, Faculty of Medicine of the University of Porto, 4200-319 Porto, Portugal; asbarros@med.up.pt (A.S.B.); ftrindade@med.up.pt (F.T.); amoreira@med.up.pt (A.L.-M.); 4LAQV/REQUIMTE, Department of Chemistry, University of Aveiro, 3810-193 Aveiro, Portugal; ritaferreira@ua.pt; 5Department of Pathology, Portuguese Oncology Institute of Porto (IPO Porto) & Porto Comprehensive Cancer Center (P.CCC), 4200-072 Porto, Portugal; 6Department of Pathology and Molecular Immunology, Institute of Biomedical Sciences Abel Salazar, University of Porto (ICBAS-UP), 4050-513 Porto, Portugal; 7Department of Clinical Pathology, Portuguese Oncology Institute of Porto (IPO Porto) & Porto Comprehensive Cancer Center (P.CCC), 4200-072 Porto, Portugal; laraujo@ipoporto.min-saude.pt

**Keywords:** prostate cancer, urine, human, biomarker, proteome, proteogenome, label-free quantitation, immunoblot

## Abstract

**Simple Summary:**

Prostate cancer (PCa) is one of the most common cancers. Due to the limited and invasive approaches for PCa diagnosis, it is crucial to identify more accurate and non-invasive biomarkers for its detection. The aim of our study was to non-invasively uncover new protein targets for detecting PCa using a proteomics and proteogenomics approach. This work identified several dysregulated mutant protein isoforms in urine from PCa patients, some of them predicted to have a protective or an adverse role in these patients. These results are promising given urine’s non-invasive nature and offers an auspicious opportunity for research and development of PCa biomarkers.

**Abstract:**

To identify new protein targets for PCa detection, first, a shotgun discovery experiment was performed to characterize the urinary proteome of PCa patients. This revealed 18 differentially abundant urinary proteins in PCa patients. Second, selected targets were clinically tested by immunoblot, and the soluble E-cadherin fragment was detected for the first time in the urine of PCa patients. Third, the proteogenome landscape of these PCa patients was characterized, revealing 1665 mutant protein isoforms. Statistical analysis revealed 6 differentially abundant mutant protein isoforms in PCa patients. Analysis of the likely effects of mutations on protein function and PPIs involving the dysregulated mutant protein isoforms suggests a protective role of mutations HSPG2*Q1062H and VASN*R161Q and an adverse role of AMBP*A286G and CD55*S162L in PCa patients. This work originally characterized the urinary proteome, focusing on the proteogenome profile of PCa patients, which is usually overlooked in the analysis of PCa and body fluids. Combined analysis of mass spectrometry data using two different software packages was performed for the first time in the context of PCa, which increased the robustness of the data analysis. The application of proteogenomics to urine proteomic analysis can be very enriching in mutation-related diseases such as cancer.

## 1. Introduction

Prostate cancer (PCa) is one of the most prevalent cancers among men and the fifth leading cause of cancer-related death [1]. When detected at early stages, PCa can be treated. However, PCa diagnosis is challenging, largely due to the low specificity of PSA tests, particularly in the diagnostic window of 4–10 ng/mL [2], which underscores the need to identify new and more accurate biomarkers.

An ideal biomarker for PCa should be non-invasively assessed, inexpensive, highly sensitive, and specific [3]. For anatomical reasons, urine is enriched in prostatic secretions and better reflects the molecular changes associated with the prostate than blood, which contains markers and confounding factors from the whole body. Urine can be serially collected, requiring minimal processing steps, and presents a simpler matrix with more stability than blood [4].

The phenotype role of proteins combined with the variety of techniques available for proteome analysis makes the search for protein markers in cancer a very attractive strategy [5]. Some promising single-protein biomarkers have been reported, such as AMBP [6] and zinc-alpha-2-glycoprotein (AZGP1) [7,8]. AMBP discriminated PCa and benign prostatic hyperplasia (BPH) patients with a highest accuracy than that estimated for PSA [9], using 2D-DIGE MALDI-TOF/TOF and immunoturbidimetry as discovery and validation approaches, respectively. AZGP1 significantly improved the prediction of PCa in a cohort of candidates for a prostatic biopsy, using isobaric stable isotope labeling and 2D-LC-MS/MS as the discovery method and Western Blot as the validation approach. Multi-marker panels have been shown to improve performance because they better reflect the cancer complexity and heterogeneity, addressing the limitations of single biomarkers. Although promising, no urine protein panel is available for clinical practice due partly to failure in clinical validation, reflecting the need to discover new biomarkers and/or new combinations of biomarkers [7,8]. Interestingly, and to the best of our knowledge, only one assay (Promark^®^) that quantifies a protein panel in prostate tissue by Mass Spectrometry (MS) is commercially available [10] and, to date, only four mRNA-based urine tests—PCA3 [11], SelectMDX [12], ExoDx Prostate(IntelliScore) [13], and MyProstateScore [14]—have been commercialized.

Cancer is driven by accumulated mutations and other genomic alterations [15]. Mutations on proteins can affect their structure, function, and stability, which may increase their susceptibility to being degraded [16]. As in other types of cancer, in PCa, a weak correlation between RNA and proteins expression is observed. Therefore, the effect of mutations should also be directly investigated at the protein level [17]. To address this inference problem, integration of genome and proteome data (proteogenome) analyses has been performed to identify mutant protein isoforms. Integrated proteogenome analysis can provide new insights into PCa pathophysiology and unveil powerful clinically applicable biomarkers. A shotgun proteomics approach combined with a mutation database has been used to detect mutated peptides related to various types of cancer, such as breast [18], colon [19], and rectal cancer [20]. Still, in PCa, it is mostly unexplored. In 2018, Kwon et al. first applied a proteogenome approach to identify six mutated peptides in the conditioned media from human PCa cell lines related to androgen-independent PCa, which are specific markers for PCa and for metastasis sites [21]. More recently, the same team identified seventy mutant peptides in PCa cell lines, of which seven were differentially expressed in PCa compared to normal tissues [22].

To identify a panel of putative protein markers to be evaluated in a non-invasively collected body fluid for PCa screening, the urine proteome and proteogenome of PCa patients were characterized by an MS-based approach. The integration of results was used to select candidate targets for small-scale clinical testing. MS is widely used to discover urinary protein biomarkers for cancer, including PCa [23]. Usually, biomarker discovery relies on a shotgun proteomics approach, followed by a validation phase using antibody-based techniques or targeted MS. Considering the complex mixture of proteins in urine, separation methodologies are important to increase sensitivity. Thus, a combination of gel-based and gel-free methods, such as GeLC-MS/MS, appears to be a robust and reproducible method for proteome analysis [24], warranting its application in the present work.

This work aims to improve the diagnosis of PCa by investigating the effect of new mutations in proteins that can be detected in urine, a non-invasively collected fluid. Additipnally, it overcomes the limitations of prior studies by using a combination of two software packages for MS data analysis, a proteogenome approach, and a detailed revision and integration of other exploratory proteome analyses to select protein targets.

## 2. Materials and Methods

### 2.1. Urine Proteome Profile of PCa Patients and Cancer-Free Subjects

#### 2.1.1. Patients and Sample Collection

Urine samples were collected, without a prior prostate massage, from patients diagnosed with PCa at the Portuguese Oncology Institute of Porto (IPO Porto, Porto, Portugal), before surgery or therapy. Patients with other types of cancer, obesity, or autoimmune diseases were excluded, and cancer-free subjects had no clinically apparent prostatic disease. All available clinical data of the subjects enrolled in this study (discovery (d) and testing cohorts) is depicted in Appendix A. The discovery cohort comprised five PCa patients and five cancer-free subjects (controls). The testing cohort comprised thirty patients and thirty cancer-free subjects, not considering benign prostate diseases, such as BPH, due to the unavailability of samples.

#### 2.1.2. Urine Sample Preparation

Urine samples were kept at 4 °C and centrifugated at 4000× *g* for 20 min at 4 °C. The supernatant (4.5 mL per sample) was collected and stored at −80 °C until laboratory analysis. Each urine sample was concentrated using a filter device (10 kDa cut-off, Vivaspin 500 Sartorius Biotech) by sequential centrifugations at 10,000× *g* for 10 min at 10 °C. Afterward, the retentate was resuspended in 0.5 M Tris pH 6.8 and 4% SDS and protein concentration were assessed by DCTM kit (Bio-Rad, Hercules, CA, USA).

#### 2.1.3. SDS-PAGE

The volume equivalent to 50 µg of protein was precipitated overnight with cold acetone (−20 °C) and centrifugated at 14,000× *g* for 30 min at 4 °C. Then, the precipitated protein was mixed 1:1 with sample Laemmli loading buffer (0.5 M Tris-HCl pH 6.8, 15% glycerol, 4% SDS, 20% 2-mercaptoethanol, bromophenol blue), heated to 100 °C for 5 min, and separated on 12% Tris-Glycine gels. Following electrophoretic separation, gels were fixed in methanol:acetic acid:water (4:1:5; for 30 min) and stained with Colloidal Coomassie Blue G250 (overnight). Gels were distained with 20% methanol until optimal contrast was achieved.

#### 2.1.4. Liquid Chromatography Tandem-Mass-Spectrometry (LC-MS/MS)

Tryptic digestion was performed according to Shevchenko et al. [25], with a few modifications. All protein bands were manually excised from the gels and sliced into ten sections. The gel pieces were washed with ammonium bicarbonate (NH_4_HCO_3_) (25 mM) and ACN (acetonitrile). Proteins were reduced with dithiothreitol (10 mM, 30 min, 60 °C) and alkylated in the dark with iodoacetamide (55 mM, 30 min, 25 °C). The gel pieces were washed with 100 mM NH_4_HCO_3_ and then with ACN. Gel pieces were vacuum-dried (SpeedVac, Thermo Savant) and proteins digested with trypsin (Thermo Scientific™, Waltham, MA, USA. Pierce™ Trypsin Protease, MS Grade) in 50 mM NH_4_HCO_3_ to a final protease: protein ratio of 1:25 (*w/w*). After 30 min on ice, 50 μL of 50 mM NH_4_HCO_3_ was added, and the samples were incubated for 16 h at 37 °C. The extraction of tryptic peptides was performed by the serial addition of 10% formic acid (FA), 10% FA:ACN (1:1) twice, and 90% ACN. Tryptic peptides were lyophilized and resuspended in 1% FA upon HPLC injection. The samples were analyzed with an Orbitrap Q Exactive (Thermo Fisher Scientific, Bremen, Germany) through the EASY-spray nano ESI source (Thermo Fisher Scientific, Bremen) that was coupled to an Ultimate 3000 (Dionex, Sunnyvale, CA, USA) HPLC system. The trap (5 mm × 300 µm inner diameter) and the EASY-spray analytical (150 mm × 75 µm) columns used were C18 Pepmap100 (Dionex, LC Packings, Sunnyvale, CA, USA), having a particle size of 3 µm. One analytical replicate was performed for each sample and blank runs were acquired between samples. For quality control of the performance of the nano-LC system, the acquisition of cytochrome C digest (1 pmol/μL) (cytochrome c digest lyophilized P/N 161089-thermo scientific) was routinely performed. Peptides were trapped at 30 μL/min in 96% of solvent A (0.1% FA). Elution was achieved with the solvent B (0.1% FA/80% acetonitrile *v/v*) at 300 nL/min. The 92 min gradient used was as follows: 0–3 min, 4 solvent B; 3–70 min, 4–25% solvent B; 70–90 min, 25–40% solvent B; 90–92 min, 40–90% solvent B; 92–100 min, 90% solvent B; 100–101 min, 90–4% solvent B; 101–120 min, 4% solvent B. The mass spectrometer was operated at 2.2 kV in the data-dependent acquisition mode. An MS2 method was used with an FT survey scan from 400 to 1600 m/z (resolution 70,000; auto-gain control target 1 × 10^6^). The 10 most intense peaks were subjected to high collision dissociation fragmentation (resolution 17,500; auto-gain control target 5 × 10^4^, normalized collision energy 28%, max. injection time 100 ms, dynamic exclusion 35 s).

#### 2.1.5. Protein Identification and Quantification

The MaxQuant (version 1.6.5.0, Thermo software) and Proteome Discoverer (version 2.2, Thermo Fisher Scientific) software packages were used for peptide identification and label-free quantification. In MaxQuant, the Andromeda, and Proteome Discoverer, the MS Amanda, and Sequest HT search engines were used to search the MS/MS spectra against the Uniprot (TrEMBL and Swiss-Prot) protein sequence database under Homo Sapiens (version December 2018). Both database search parameters were as follows: methionine oxidation, protein N-term acetylation and phosphorylation, as variable modifications, and cysteine carbamidomethylation as a fixed modification. The mass tolerance of precursor mass was 20 ppm for MaxQuant and 10 ppm for Proteome Discoverer, and fragment ion mass tolerance was 0.15 Da (MaxQuant) and 0.02 Da (Proteome Discoverer). Minimal peptide length was set to 7 amino acids and, at most, 2 missed cleavages were allowed for both software. The false discovery rate (FDR) for identification was set to 1% at peptide and protein levels. Only the top-ranking protein of each group (master proteins), identified with at least two peptides, were considered. Exclusion of contaminants relied on those identified by the MaxQuant software and the cRAP protein sequences—THE GPM (https://www.thegpm.org/crap/) (accessed on 2 April 2019).

The MS proteome data have been deposited on the ProteomeXchange Consortium via the PRIDE [26] partner repository with the data set identifier PXD017902.

#### 2.1.6. Exploratory Analysis of Urine Proteome Data

The protein abundances in Proteome Discoverer (normalized to the respective median) and normalized LFQ intensities in MaxQuant were log 2-transformed. In an exploratory analysis of proteome data, the proteins identified in all individuals were used as variables to perform Principal Component Analysis (PCA) and Heatmap analyses. These analyses were performed on MetaboAnalyst 5.0 [27]. To identify dysregulated proteins in PCa patients, the fold-change in protein abundance between PCa patients and cancer-free subjects was then calculated from the average log2 difference of protein intensities. Student’s *t*-test assessed the statistical significance of this difference.

#### 2.1.7. Comparison with a Previous Bioinformatic Analysis of Putative Urinary Markers of PCa and Selection of Candidate Protein Targets for the Testing Phase

Dysregulated proteins were compared with the results of a bioinformatic analysis focused on comparing and mining the proteome profile of tumor prostate tissue and urine from PCa patients reported by several MS studies [28]. The bioinformatic analysis reported 2641 and 616 dysregulated proteins in tumor prostate tissue and urine from PCa patients, respectively. To place urine proteome as a reflection of events taking place in prostate tissue and to identify specific urinary protein targets for PCa, the dysregulated proteins identified in tumor prostate tissue and urine from PCa patients were compared, resulting in 339 overlapping proteins. In this sense, the dysregulated proteins identified by MS in the present work, common to the 2641 dysregulated proteins expressed in tumor prostate tissue or to the 339 urinary proteins with prostate expression, correspond to the selection criteria of candidate proteins to be tested. Then, the selected proteins were compared with the normal human urinary proteome [29].

#### 2.1.8. Measurement of Candidate Protein Targets in Urine Using Immunoblot

The selected protein targets from the discovery phase were tested by slot blot or Western blot immunoassays. In slot blot analysis, performed according to Caseiro et al. [30], the urine protein concentrated fraction was diluted in TBS to a final protein concentration of 0.01 μg/μL and slot-blotted onto a nitrocellulose membrane (Amersham Protran NC 0.45; Amersham Pharmacia Biotech, Buckinghamshire, UK). Antibodies specificity, selectivity, and sensitivity were assessed previously through Western blot by the bands appearing at the expected molecular weights without evidence of non-specific binding of the antibodies. The blocking and incubation conditions were optimized as follows: EFEMP1 (GTX111657: 1:1000, 1 h; GE Amersham-NA934: HRP-linked donkey anti-rabbit 1:10,000); AMBP (sc-81948: 1:1000, 1 h; GE Amersham-NA931: HRP-linked sheep anti-mouse 1:5000); LMAN2 (sc-130026, 1 h; GE Amersham-NA931: HRP-linked sheep anti-mouse 1:5000). Regarding Western blot, 20 µg of protein from each sample was separated on a 12% SDS-PAGE gel and transferred onto nitrocellulose membranes. In both immunoblot experiments, Ponceau S staining was used to normalize the antibody signal to total protein levels. In any case, the membranes were washed with TBS-T (TBS 25 Mm Tris−HCl, pH 7.4, 150 Mm NaCl, 0.1% Tween 20) and imaged in a ChemiDocTM Touch imaging system (Bio-Rad) using the Enhanced Chemiluminescence kit (ECL Select Western Blotting Detection Reagent, RPN2235, Amersham). Optical density was assessed with Image Lab Software (Bio-Rad) and normalized to a loading control sample. Western blot conditions were: CDH1 (GTX629691: 1:1000, 1 h; GE Amersham-NA931: HRP-linked sheep anti-mouse 1:5000); TTR (GTX100577: 1:500, 1 h; GE Amersham-NA934: HRP-linked donkey anti-rabbit 1:10,000).

#### 2.1.9. Measurement of Urinary PSA Levels

Urinary PSA levels were determined using the same method (Elecsys total PSA, 08791732500) used to determine serum PSA levels. This electrochemiluminescence assay is used in the clinical routine of IPO Porto. It quantifies total PSA (free + complexed PSA) using a Cobas e 801 module, a member of Roche Cobas 8000 Modular Analyzer (Roche, Woerden, The Netherlands).

### 2.2. Urine Proteogenome Profile of PCa Patients and Cancer-Free Subjects

#### 2.2.1. Identification of Cancer-Associated Mutations

Considering the high impact of mutations on cancer progression, the proteogenome profile of urine from PCa patients was explored. For this, mass spectra resulting from the MS analysis were searched against a database built into the Pinnacle software (https://rimuhc.ca/-/protein-quantification-software-pinnacle?redirect=%2Fproteomics-software, accessed on 5 January 2022). This type of analysis aimed to investigate the existence of cancer-associated mutations that were translated in proteins present in the urine from PCa patients. To select high-confidence urinary proteins with a very likely origin in the prostate, only mutations on proteins present in all samples and with known prostate expression were considered. The prostate proteome was searched in the HPA database and in the above-mentioned bioinformatic analysis [28]. The prostate proteome in the HPA consisted of proteins with evidence at the protein level and its last access was on 8 November 2021.

#### 2.2.2. Exploratory Analysis of Urine Proteogenome Data

The abundances of proteins with known prostate expression in Pinnacle were log 2-transformed. In an exploratory analysis of proteogenome data, the levels of mutant protein isoforms identified in all individuals were used as variables to perform Principal Component Analysis (PCA) and Heatmap analyses. These analyses were performed on MetaboAnalyst 5.0 [27]. To identify dysregulated proteins with mutations in PCa patients, the fold-change in protein abundance between PCa patients and cancer-free subjects was then calculated from the average log2 difference of protein intensities. Student’s *t*-test assessed the statistical significance of this difference.

#### 2.2.3. Integration with the Cancer Genome Atlas (TCGA), DisGeNET and Literature Data

To investigate whether mutations identified in proteins with known prostate expression were already described in PCa, TCGA, DisGeNET (v7.0), and literature data were searched.

TCGA is a cancer genomics consortium that generates data (https://www.cancer.gov/tcga, accessed on 12 January 2022) encompassing the profiling of over 20,000 primary tumors and matched non-tumoral samples related to various human cancers, including PCa. The characterization of PCa samples disclosed 20,237 mutated genes and 33,334 mutations. DisGeNET is one of the largest repositories of Gene-Disease (GDA) and Variant-Disease (VDA) Associations [31]. The latest version of DisGeNET contains 1,134,942 GDAs and 369,554 VDAs. In the present work, variants associated with PCa were extracted from the Prostate Carcinoma C0600139 (January 2022).

#### 2.2.4. Comparison of the Levels of Native and Mutant Forms of Proteins in the Urine from PCa Patients

To investigate the influence of mutations on the abundance of proteins with known expression in the prostate, the levels of their native and mutant forms were compared.

#### 2.2.5. Prediction of the Likely Impact of Single-Residue Substitutions in Proteins

The PolyPhen-2 (Polymorphism Phenotyping v2) web tool was used to predict the likely impact of each amino acid substitution on the structure and function of the proteins with known prostate expression [32]. Each mutation is assigned a score, which is the probability of the substitution being damaging, in addition to a sensitivity and specificity value of the prediction confidence. According to the PolyPhen-2 tool, single-residue substitutions in the protein sequence can be classified as benign (score: 0–0.4), possibly damaging (score: 0.4–0.9), or probably damaging (score: 0.9–1) [33].

#### 2.2.6. Protein–Protein Interaction Analysis

Due to the pivotal role of Protein–Protein interactions (PPIs) in cancer and the possible effect of mutations on its dynamics, the interactions between proteins in which point mutations has been identified were explored. For this, the STRING database v 11.5 was sourced on 12 January 2022, and only protein interactions with a confidence score of ≥0.4 were considered [34]. However, we must be cautious when extrapolating the significance of these PPIs to biological fluids such as urine, as most PPIs are identified or predicted from studies in cells and tissues.

#### 2.2.7. Prediction of the Likely Impact of Single-Residue Substitutions in Protein–Protein Affinity

Considering the impact of mutations on PPIs, the SAAMBE-SEQ Web Server was used to predict the effect of point mutations detected in this work on protein binding affinity [35].

### 2.3. Statistical Data Analysis

Statistical analyses were carried out in R software for Windows version 3.6.2 and GraphPad Prism version 6.0 (GraphPad Software, Inc.; San Diego, CA, USA). The Shapiro normality test and visual inspection of the histograms were used to assess the data distribution. To evaluate the effect size of the dysregulated proteins when comparing the tested groups, Cohen’s d was determined. Differences were considered statistically significant if *p*-value was ≤ 0.05. The clinical parameters and protein levels are expressed as mean ± standard deviation (SD).

## 3. Results

### 3.1. Urine Proteome Profile of PCa Patients and Cancer-Free Subjects

To identify potential protein targets for PCa prediction, shotgun proteomics was performed in urine collected from PCa patients and cancer-free subjects. To boost MS data analysis, a combination of two different software packages, MaxQuant and Proteome Discover, sourcing three databases (Andromeda, Amanda, and Sequest HT) in total, was used.

Considering only the top-ranking protein of each group identified with at least two peptides and filtering out identifications from reversed sequences and contaminants, 605 and 592 urinary proteins were identified by MaxQuant and Proteome Discoverer, respectively. In total, 732 proteins were identified, excluding those common to both software.

#### 3.1.1. Exploratory Analysis of Urine Proteome Data

Aiming to select and identify proteins of interest for PCa monitoring, only proteins present in all samples analyzed by MaxQuant (82 proteins) and by Proteome Discoverer (84 proteins) were considered for further analysis. These high-confidence proteins were separately used for Principal Component Analysis (PCA) (Figure 1A and Figure 2A) and Heatmap analyses (Figure 1B and Figure 2B). In both software, no separation of groups was observed in the PCa analysis. However, the proteins identified by the MaxQuant software alone seem to provide a discrimination between PCa patients and non-cancer subjects based on two protein clusters, depicted in the heatmap: AZGP1(zinc-alpha-2-glycoprotein)-SPP1 (Osteopontin); CD14 (Monocyte differentiation antigen CD14)-MASP2 (Mannan-binding lectin serine protease 2) (Figure 1B). In the first cluster, proteins are mostly upregulated in PCa patients compared to non-cancer subjects, while in the second cluster proteins are predominantly downregulated in PCa patients.

Then, differential protein analysis revealed 18 dysregulated proteins in PCa, with 4 proteins (*p*-value ≤ 0.05) identified only by Proteome Discoverer, 9 proteins only by MaxQuant analysis, and 5 proteins (Cadherin-1 (CDH1), EGF-containing fibulin-like extracellular matrix protein 1 (EFEMP1), Prostate-specific antigen (PSA) (KLK3), Secreted and transmembrane protein 1 (SECTM1), and Transthyretin (TTR)) discovered by both software. Altogether, 11 proteins were significantly downregulated (fold change less than 1), and 7 proteins were significantly upregulated (fold change greater than 1) in PCa patients (Table 1 and Table 2). Reassuringly, the most widely used biomarker for PCa diagnosis, PSA, was one of the dysregulated proteins in common in the analysis by both software packages. When the tested groups were compared, proteins showing significant differences (*p*-value ≤ 0.05) and revealed a “large” effect-size (|Cohen’s d|) > 0.8 (Table 1 and Table 2). Besides a large effect-size, dysregulated proteins identified by both software presented a consistent direction of dysregulation. It is noteworthy that in the heatmap of MaxQuant data, seven proteins (TTR, KLK3, SECTM1, CDH13, AMY2A, EFEMP1, ITIH4, HSPG2, PTGDS, CDH1, and LMAN2) responsible for the separation of groups were also found dysregulated in PCa patients. It was observed that the decreased levels of SECTM1, CDH13, AMY2A, EFEMP1, ITIH4, HSPG2, PTGDS, CDH1, and LMAN2 and increased levels of TTR and KLK3 characterized the urine proteome of PCa patients.

#### 3.1.2. Comparison with a Previous Bioinformatic Analysis of Putative Urinary Markers of PCa and Selection of Candidate Protein Targets for the Testing Phase

To select the most promising proteins for further analysis, dysregulated proteins revealed by MS analysis were compared with proteins resulting from a bioinformatic analysis integrating urine and tumor tissue proteomes of PCa from several MS studies [28]. From this comparison, some common proteins emerged, such as AMBP, CDH1, EFEMP1, KLK3, SECTM1, LMAN2, and TTR.

From the previous study of our group, the dysregulated proteins AMBP, KLK3, LMAN2, and TTR were found dysregulated in urine and tumor tissue from PCa patients, while SECTM1 was only found in urine from PCa patients, and CDH1 and EFEMP1 were only in PCa tissue.

Taken together, and keeping in mind that candidate targets should be urinary proteins with prostate expression, AMBP, CDH1, EFEMP1, KLK3, LMAN2, and TTR were selected for testing in an independent cohort. The presence of these proteins in the urine was already expected, because they are characteristic of the normal human urine proteome [29].

#### 3.1.3. Measurement of Candidate Protein Targets in Urine

Five protein targets, AMBP, CDH1, EFEMP1, LMAN2, and TTR were selected for immunoblot-based testing in a larger and independent cohort (testing group). However, none of the MS findings could be reproduced (Appendix A, Appendix A). Measurement of urinary PSA levels in the testing cohort did not agree with the MS findings (*p* = 0.29, Mann–Whiney test). The results are shown in Figure 3.

### 3.2. Urine Proteogenome Profile of PCa Patients and Cancer-Free Subjects

#### 3.2.1. Identification of Cancer-Associated Mutations

To characterize the proteogenome landscape of urine from PCa patients, MS/MS spectra were searched against a repository of information from a wide variety of databases encompassing somatic mutations. This search resulted in identifying 6418 mutated peptides corresponding to 1665 mutant protein isoforms. Of these, 609 mutated peptides, which correspond to 417 mutant protein isoforms, were associated with cancer. Only mutant protein isoforms that occurred in all urine samples (322 proteins) were selected for further analysis. Immunoglobulins and highly abundant urinary proteins (serum albumin, uromodulin, serotransferrin) were excluded due to their high abundance in biological samples and the lack of specificity for cancer, resulting in 170 proteins. These 170 proteins corresponded to 122 proteins after filtering out duplicates. As our focus was high confidence proteins with mutations whose origin was very likely the prostate, these data were integrated with the prostate proteome searched in the HPA database and in a bioinformatic analysis [28], resulting in 86 proteins with known expression in the prostate (Appendix A). Among these proteins are some of known relevance for PCa, namely Acid ceramidase (ASAH1), Extracellular superoxide dismutase [Cu-Zn] (SOD3), Glutathione S-transferase P (GSTP1), Osteopontin (SPP1), Prostatic acid phosphatase (PAP), and Zinc-alpha-2-glycoprotein (ZAG).

#### 3.2.2. Exploratory Analysis of Urine Proteogenome Data

The levels of the mutant protein isoforms were used for PCA (Principal Component Analysis) (Figure 4A) and Heatmap analyses (Figure 4B). No group separation was observed in the PCA of the proteogenome profile of PCa patients. However, the heatmap indicates a discrimination between PCa patients and non-cancer subjects based on two protein clusters: ITIH4*G893S (Inter-alpha-trypsin inhibitor heavy chain H4)-LMAN2*D222N (Vesicular integral-membrane protein VIP36); KLK3*C209Y (PSA)-MVB12B*T198M (Multivesicular body subunit 12B) (Figure 4B). In the first cluster, mutant forms of proteins are mostly downregulated in PCa patients compared to non-cancer subjects, while in the second cluster mutant forms of proteins are upregulated predominantly in PCa patients.

#### 3.2.3. Integration with the Cancer Genome Atlas (TCGA), DisGeNET and Literature Data

According to TCGA, DisGeNET, and the literature, only three of the mutations identified in the 86 proteins with known prostate expression have already been described. These mutations (rs17632542, rs1695, rs7041) were mapped on KLK3 (PSA) [36], GSTP1 (Glutathione S-transferase P) [37,38], and GC (Vitamin D-binding protein) [39], respectively. To the best of our knowledge, there is no association of the remaining mutant protein isoforms with PCa. Especially notable are the proteins SPP1, VASN, ASAH1, RBP4, and ASS1, which, until now, have had no mutation related to PCa described in the literature.

#### 3.2.4. Comparison of the Levels of Native and Mutant Forms of Proteins in the Urine from PCa Patients

The analysis of proteogenome data revealed 6 differentially abundant mutant protein isoforms in PCa patients compared with cancer-free individuals, namely Protein AMBP (AMBP*A286G), Sodium/hydrogen exchanger 9B1 (SLC9B1*N70S), Basement membrane-specific heparan sulfate proteoglycan core protein (HSPG2*Q1062H), Zinc finger protein 624 (ZNF624*S207F), Vasorin (VASN*R161Q), and Complement decay-accelerating factor (CD55*S162L) (Appendix A, Appendix A). Mutant AMBP isoform was upregulated in PCa patients, while the remaining 5 differentially abundant mutant protein isoforms were downregulated.

Comparing the proteome profile analysis of MaxQuant and Proteome Discoverer with the proteogenome profile of PCa patients resulted in 30 and 31 common proteins, respectively. Of these common proteins, AMBP, CDH1, EFEMP1, HSPG2, ITIH4, KLK3, LMAN2, PTGDS, VASN, and CD55 proteins stood out. The native form of AMBP, CDH1, EFEMP1, HSPG2, ITIH4, KLK3, LMAN2, and PTGDS proteins was found dysregulated in urine from PCa patients, but only the mutant protein isoforms (AMBP*A286G; HSPG2*Q1062H) were found dysregulated (Appendix A). In the remaining common proteins, the presence of mutations did not affect their abundance in urine. The native form of VASN and CD55 proteins was not found dysregulated in the urine from PCa patients, but their mutant protein isoforms (VASN*R161Q; CD55*S162L) were.

The mutations identified in these proteins and in those with recognized relevance to PCa are summarized in Table 3.

#### 3.2.5. Prediction of the Likely Impact of Single-Residue Substitutions in Proteins

With the purpose of determining the potential impact of point mutations on protein function, PolyPhen-2 tool was used. It is worthy of mention that AMBP*A286G and CD55*S162L mutant protein isoforms were predicted to be probably damaging, while SLC9B1*N70S, ZNF624*S207F, VASN*R161Q, and HSPG2*Q1062H were predicted to be benign. Most point mutations were predicted to be possibly or probably damaging. The results are presented in Table 4 and Appendix A.

#### 3.2.6. Protein–Protein Interaction Analysis

In addition to impacting the function of proteins, mutations can also affect interactions between proteins and, consequently, important biological processes and signaling pathways. To predict interactions between the proteins in which point mutations were identified, the STRING search tool was used. As shown in Figure 5, the network consisted of 86 connected proteins (nodes) through 214 edges with different confidence levels. The protein–protein interaction enrichment *p*-value was <1.0 × 10^−16^. Reactome enrichment analysis showed 12 pathways enriched in this network (Appendix A). Regulation of Insulin-like Growth Factor (IGF) transport and uptake by Insulin-like Growth Factor Binding Proteins (IGFBPs) was the third most important pathway in this network, while Extracellular matrix (ECM) organization was the tenth. This network shows predicted interactions between most of the proteins.

#### 3.2.7. Prediction of the Likely Impact of Single-Residue Substitutions in Protein–Protein Affinity

To predict the impact of point mutations on PPIs, the SAAMBE-SEQ tool was used. The likely effect of AMBP*A286G, HSPG2*Q1062H, VASN*R161Q, and CD55*S162L point mutations on protein–protein interactions was scrutinized. Point mutations detected on SLC9B1 and ZNF624 were not examined as these proteins do not interact with any proteins in the network. Additionally, the impact of point mutations on proteins involved in the IGF pathway was also explored. This analysis revealed that the likely effect of these point mutations is destabilizing for PPIs (Appendix A).

## 4. Discussion

The limitations and the invasive nature of serum PCa screening have driven the discovery of new candidate urinary biomarkers, especially protein markers. However, so far, none has translated into clinically useful tools, reflecting the need to discover novel biomarkers and/or new combinations of biomarkers. Thus, this study aimed to take advantage of a non-invasively collected biofluid, urine, and a high throughput approach, proteomics, to identify new protein targets for predicting the risk of developing PCa. This work was divided into three stages: characterization of the urine proteome profile and selection of protein targets; testing of shortlisted protein targets in a larger, independent cohort; and characterization of the urine proteogenome profile. The urine proteome profile of PCa and cancer-free subjects was analyzed by two software packages and 18 dysregulated proteins, of which 5 (TTR, EFEMP1, CDH1, SECTM1, KLK3) common to both software, were found. The integration of the urine proteome profile of PCa patients with proteome data from other studies reviewed by us [28] supported the selection of potential discriminatory protein targets. As a result, AMBP, CDH1, EFEMP1, LMAN2, and TTR stood out as potential targets and were tested in an independent cohort of patients. In this testing phase, incubation with anti-E-cadherin did not result in a band around 120 kDa (full-length protein), but rather a band about 80 kDa. We realized that this 80 kDa fragment corresponded to soluble E-cadherin (sE-cadherin) and has been previously identified in tissue and serum from PCa patients [93,94] and in urine from patients with other cancers [95,96], using antibody-based techniques. Concerning PCa, as far as we know, here we present the first report of the detection of sE-cadherin fragment in the urine. Kuefer et al. [93] suggested that the 80 kDa fragment is originated from the extracellular domain of full-length E-cadherin. Increased levels of sE-cadherin have been reported in serum and tumor prostate tissue from PCa patients and are correlated with disease stage [94,97,98]. Differential abundances of these MS-detected proteins were tested in an independent cohort using immunoblot, but different variations were observed. Additionally, urinary PSA levels were also assessed in this independent cohort, but did not distinguish PCa patients from controls, which agrees with other studies [99].

The proteogenome landscape of urine from PCa patients was then characterized and 1665 mutant protein isoforms were disclosed, of which 417 were cancer-related mutations. After considering only mutations present in all urine samples and proteins with known prostate expression, 86 mutant protein isoforms emerged. Among these proteins are some of known relevance for PCa, namely Acid ceramidase (ASAH1), Extracellular superoxide dismutase [Cu-Zn] (SOD3), Glutathione S-transferase P (GSTP1), Osteopontin (SPP1), Prostatic Acid Phosphatase (PAP), and Zinc-Alpha-2-Glycoprotein (ZAG). PAP is gaining renewed interest due to its superior predictive role of cause-specific survival and GS compared to serum PSA in men with high risk PCa [100,101]. Remarkably, it was recently suggested that a form of PAP (PLPAcP) associates with early PCa [102]. Identifying a new mutation in this protein in a non-invasive biological fluid, adding to the prediction of PAP mutation to be probably damaging, strengthens the renewed interest in its study in PCa. Mutations found on the 86 proteins were searched for in databases and the literature and, to the best of our knowledge, only rs17632542 [36,103,104,105], rs1695 [37,38,106,107], and rs7041 [39] mutations mapped on PSA, GSTP1, and GC proteins have been described in the PCa context. In that vein, these results validate the proteogenome analysis performed in the present study.

The analysis of the urine proteogenome profile of PCa patients revealed 6 differentially abundant mutant protein isoforms, namely AMBP*A286G, SLC9B1*N70S, HSPG2*Q1062H, ZNF624*S207F, VASN*R161Q, and CD55*S162L. From the comparison of the proteome and proteogenome profile of PCa patients, AMBP, CDH1, EFEMP1, KLK3, and LMAN2 proteins stood out. Their native form was found dysregulated in urine from PCa patients, but the same was not observed with their mutant form, with the exception of AMBP*A286G and HSPG2*Q1062H. These results may explain the differences between MS and immunoblot data, because the antibodies either do not recognize the mutated peptides or do not specifically recognize them.

PPIs play a pivotal role in most biological processes. Dysregulation of these protein interactions may result in pathological conditions, such as cancer, being involved in tumor progression, invasion, and metastasis [108,109]. In this sense, PPIs have been claimed as promising therapeutic targets for numerous types of cancer, including for PCa. For this type of cancer, 28 small molecules and 14 peptides have been proposed to disrupt PPIs with relevance to PCa progression [110]. To explore PPIs between proteins with known prostate expression and the pathways in which these interactions were involved, the STRING tool was used. In this analysis, the IGF transport and uptake by IGFBPs proved to be the third most important pathway in the network. The IGF axis is a network of ligands (GF1, IGF2, insulin) and IGFBP receptors (IGF1R, IGF2R, INSR), the latter being responsible for mediating the activity of IGFs [111]. IGFs are oncogenic regulators, promoting prostate tumor growth, survival, and proliferation, and the role of IGF axis has been well documented in PCa. For instance, IGFBP-2 enhanced proliferation of androgen-independent prostate cancer cells [112] and IGF-I levels were found raised in serum and prostate tissue from PCa patients, being a predictor of risk for this type of cancer [113,114]. In accordance with this, IGF1R and INSR act as oncogenes in PCa, enhancing tumor growth, proliferation, invasion, and angiogenesis [115]. Considering the relevance of the IGF pathway in PCa, the impact of mutations on the interaction of proteins involved in this pathway was predicted. According to SAAMBE-SEQ, the mutations were predicted to destabilize all PPIs involved in the IGF pathway, which naturally could affect this pathway and consequently the progression of PCa.

To investigate the likely impact of each amino acid substitution on protein function and PPIs involving the dysregulated mutant protein isoforms (AMBP*A286G, SLC9B1*N70S, HSPG2*Q1062H, ZNF624*S207F, VASN*R161Q, and CD55*S162L), the PolyPhen-2 and SAAMBE-SEQ prediction tools were used. The role of the SLC9B1 and ZNF624 proteins on cancer is completely unknown, so the downregulation of their mutant protein isoforms and the prediction of their benign impact do not allow conclusions to be drawn. HSPG2, in its intact form, is a well-described pro-angiogenic molecule, being correlated with GS and increased cell proliferation and viability [55,56,116]. The intact form of this protein was found increased in tumor prostate tissue, but in sera from PCa patients raised levels of HSPG2-derived fragments resulting from matrix metalloproteinase 7 (MMP7) degradation were observed. These fragments were mostly originated from domain IV and were not present in sera from non-cancer subjects, suggesting that HSPG2 cleavage occurs during metastasis and before the protein enters the bloodstream. Using an in silico analysis, Grindel et al. predicted that domains III and V of HSPG2 are the most prone to cleavage by MMP-7 and generate new peptides for other extracellular proteases to digest [55]. Curiously, in this work, the mutated peptide identified in the mutant HSPG2 isoform is located on domain III. The cleavage of HSPG2 and other components of basement membrane occurs during PCa cell invasion and is orchestrated by proteases such as MMPs, cathepsin L, and BMP1/Tolloid-like proteases. Both Cathepsin L and BMP1/Tolloid-like proteases cleave HSPG2 in domain V, originating the Endorepellin [117] and LG3 [118] peptides, respectively. Unlike the intact form, cleaved Endorepellin and LG3 peptides behave as powerful anti-angiogenic factors, being claimed as potential therapeutic targets for cancer [118]. In fact, the administration of endorepellin to mice with squamous cell carcinomas and lung carcinomas resulted in mitigation of tumor growth, angiogenesis and metabolism and promotion of tumor hypoxia [119]. Accordingly, LG3-diminished levels were noticed in breast cancer cells and in plasma from breast cancer patients [120]. Only the LG3 peptide has been detected in urine [121,122]. In PCa, both the existence and the role of these peptides are unknown, and the only recognized HSPG2 protease is MMP7. A complex network between HSPG2 and other basement membrane components, such as collagens, laminin, and nidogen is responsible for ECM integrity. When this integrity is disturbed, the metastatic process is compromised [123]. In the present work, mutations were identified in HSPG2, collagens, nidogen, and in other proteins involved in ECM organization. When the impact of these mutations on PPIs was predicted, they all proved to be destabilizing, which eventually affects ECM dynamics and tumor progression. All these results, together with the fact that the HSPG2*Q1062H point mutation was predicted to be benign and the mutant peptide was downregulated in PCa patients, suggest that this mutant peptide may have beneficial effects in patients with PCa and opens doors for its study in PCa treatment. Concerning the AMBP protein, it is cleaved into three chains, namely Alpha-1-microglobulin, Bikunin, and Trypstatin. The function of the AMBP protein in cancer remains undisclosed. However, it has been claimed that the AMBP-derived product bikunin is underexpressed in oral squamous cell carcinoma and plays an antitumor role [40]. In line with this, there is evidence that bikunin significantly prevented tumor invasion and metastasis in Lewis lung carcinoma and ovarian carcinoma cells [124,125]. Curiously, in this work, the mutant peptide identified in the AMBP isoform is located on the bikunin fragment. The mutation identified in AMBP was predicted to be probably damaging, destabilized all PPIs in which AMBP was involved, and resulted in an upregulation of mutant AMBP isoform in PCa patients. This may suggest a detrimental role of this mutation on PCa patients. Regarding CD55, it blocks complement response by accelerating the decay of C3 and C5 convertases [126] and is involved in PCa cell survival and metastasis [92]. This interplay between CD55 and C3 is visible by their interaction in the STRING network. The mutation detected on the CD55 protein was predicted to be probably damaging and destabilizing for CD55-C3 interaction. With these findings, it seems reasonable to suspect the detrimental role of this mutation on PCa patients. Regarding VASN, it is a known inhibitor of TGF-β signaling [127]. The TGF-β pathway has a dual role in cancer, because it prevents cell proliferation in early stages and in advanced stages stimulates proliferation, epithelial-to-mesenchymal transition (EMT) and evasion of immune surveillance, and attenuates apoptosis [128]. The mechanism involved in this inhibitory action of VASN on TGF-beta was revealed in breast cancer cell lines. It was demonstrated that a soluble form of VASN resulting from the proteolytic shedding of its extracellular domain by Metalloprotease domain 17 (ADAM17) is responsible for controlling the TGFβ pathway [129]. In PCa, the role of VASN is largely unexplored, including the interplay between the VASN and TGFβ pathways. However, overexpression of VASN in prostate tumor tissue and in serum from PCa patients and the subsequent promotion of cell proliferation and PCa progression have already been reported, in agreement with other types of cancer [90]. Interestingly, in this work, the mutated peptide identified in the VASN protein is located on the extracellular domain of the protein, the domain cleaved by ADAM17. The mutation identified in VASN resulted in a downregulation of this mutant protein isoform in PCa patients and was predicted to be benign, which may suggest a protective role of this mutation on PCa patients.

These findings indicate that, in mutational diseases such as cancer and in biofluids with high proteolytic activity, such as urine, the application of proteogenomics to urine analysis and the study of peptides can be very enriching because point mutations can go unnoticed at the protein level but are detected at the peptide level. This may sharpen or renew interest in underexplored targets, as observed in this work. We hope to address some of these questions in future work. Furthermore, it would be interesting to test these mutant peptides by an MS-targeted approach such as MRM, but this is beyond the scope of this work. This work’s novelty lies in the proteogenome characterization of urine from PCa patients and the combined analysis of MS data using two different software packages, increasing certainty in the identification of urinary proteins modulated by PCa.

## 5. Conclusions

The majority of mutations identified in this work have never been associated with PCa, and some are predicted to be damaging, which offers an auspicious opportunity for research and development of PCa biomarkers, especially in the HSPG2 context. Additionally, the discovery of cancer-associated mutations in PCa-related proteins in urine is promising given this biofluid’s non-invasive and dynamic nature.

## Figures and Tables

**Figure 1 cancers-14-02001-f001:**
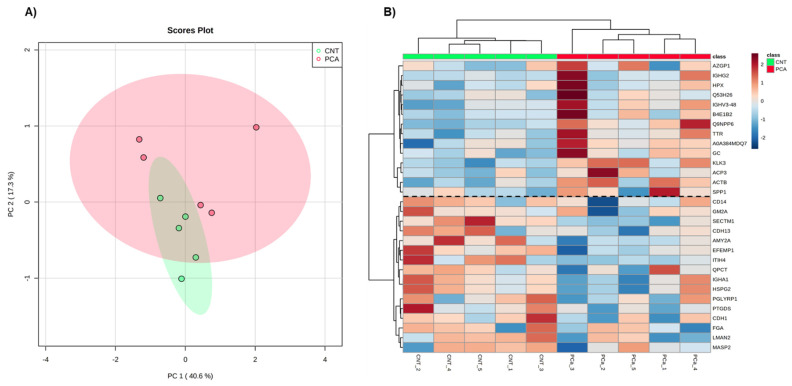
Exploratory analysis of proteome data from MaxQuant. (**A**) Principal Component Analysis of the urine proteome of the two groups. (**B**) The heatmap of proteins identified in all individuals. Samples are represented in columns and proteins in rows. Proteins whose gene name is not available are indicated by their UniProt accession number. The dashed line on the heatmap indicates the two clusters of proteins.

**Figure 2 cancers-14-02001-f002:**
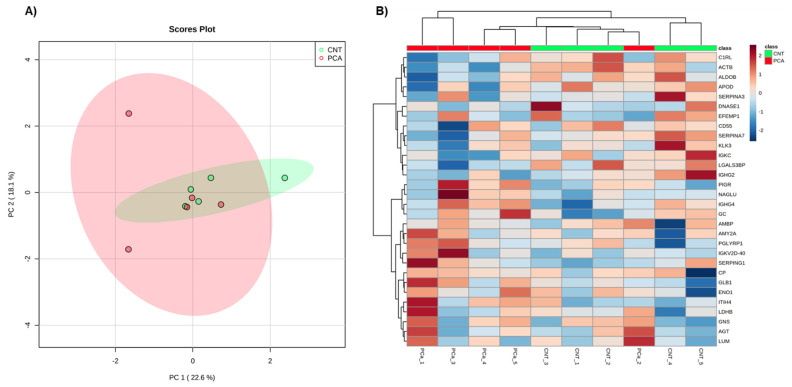
Exploratory analysis of proteome data from Proteome discoverer. (**A**) Principal Component Analysis of the urine proteome of the two groups. (**B**) The heatmap of proteins identified in all individuals. Samples are represented in columns and proteins in rows.

**Figure 3 cancers-14-02001-f003:**
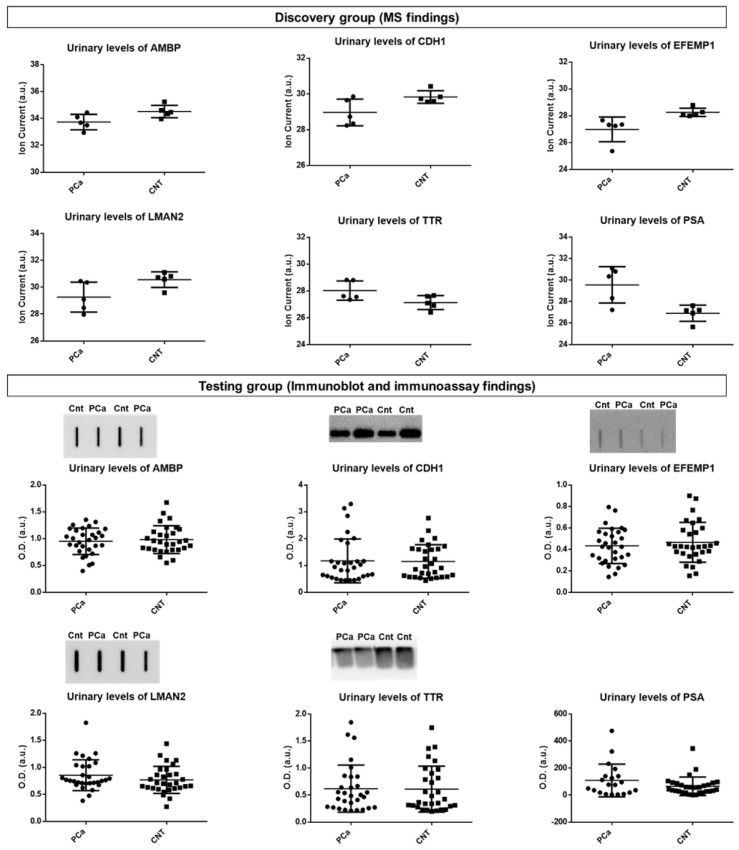
Urinary protein levels of the candidate targets for PCa in the discovery group (using MS) and in the testing group (using immunoblot and immunoassay). MS: mass spectrometry.

**Figure 4 cancers-14-02001-f004:**
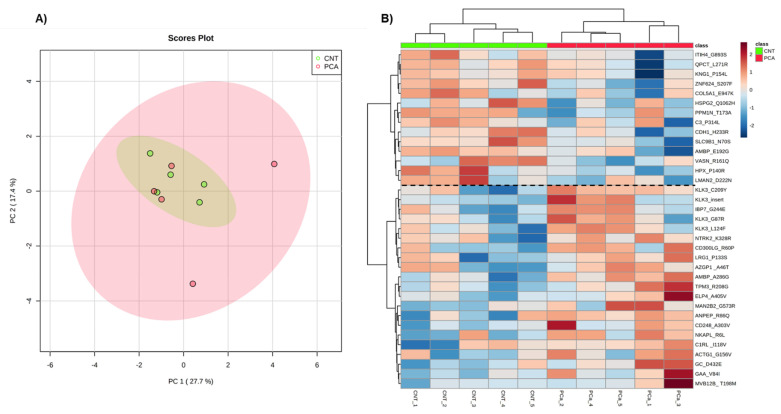
Exploratory analysis of proteogenome data from Pinnacle. (**A**) Principal Component Analysis of the urine proteogenome of the two groups. (**B**) The heatmap of mutant proteins identified in all individuals. Samples are represented in columns and proteins in rows. Proteins are identified by their gene name, and the mutation identified. The dashed line on the heatmap indicates the two clusters of proteins.

**Figure 5 cancers-14-02001-f005:**
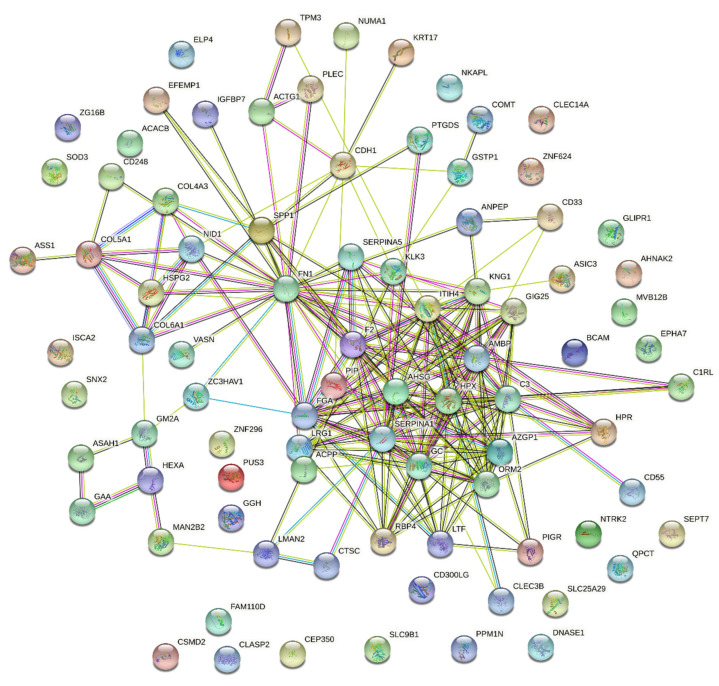
PPI network of 86 mutated proteins with known expression in the prostate.

**Table 1 cancers-14-02001-t001:** Dysregulated proteins between PCa patients and cancer-free subjects (Proteome Discoverer).

Uniprot ID	Protein Name	Gene Name	*p*-Value	Cohen’s d [Lower; Upper 95% CI]
P07288	Prostate-specific antigen	KLK3	0.00	4.21 (3.50; 4.91)
Q8WVN6	Secreted and transmembrane protein 1	SECTM1	0.01	−2.16 (−2.39; −1.93)
P12830	Cadherin-1	CDH1	0.03	−1.73 (−2.05; −1.41)
P0DOX5	Immunoglobulin gamma-1 heavy chain	N/A	0.03	1.73 (1.39; 2.07)
Q12805	EGF-containing fibulin-like extracellular matrix protein 1	EFEMP1	0.03	−1.68 (−2.25; −1.12)
P02766	Transthyretin	TTR	0.03	1.66 (0.86; 2.46)
P01861	Immunoglobulin heavy constant gamma 4	IGHG4	0.04	1.52 (0.90; 2.15)
P01034	Cystatin-C	CST3	0.05	1.50 (0.91; 2.08)
Q01459	Di-N-acetylchitobiase	CTBS	0.05	−1.44 (−1.86; −1.02)

The protein identification and label-free quantification performed by the Proteome Discoverer software revealed nine dysregulated proteins (*p*-value ≤ 0.05) between the tested groups. These proteins are shown in this table along with their *p*-value and effect size. The Cohen’s d for individual proteins is presented together with the lower and upper 95% confidence interval (CI). Abbreviation: Confidence interval (CI).

**Table 2 cancers-14-02001-t002:** Dysregulated proteins between PCa patients and cancer-free subjects (MaxQuant).

Uniprot ID	Protein Name	Gene Name	*p*-Value	Cohen’s d [Lower; Upper 95% CI]
Q8WVN6	Secreted and transmembrane protein 1	SECTM1	0.01	−2.10 (−2.48; −1.73)
P07288	Prostate-specific antigen	KLK3	0.01	2.01 (1.08; 2.95)
P41222	Prostaglandin-H2 D-isomerase	PTGDS	0.01	−1.97 (−2.44; −1.49)
Q14624	Inter-alpha-trypsin inhibitor heavy chain H4	ITIH4	0.01	−1.96 (−2.32; −1.60)
Q12805	EGF-containing fibulin-like extracellular matrix protein 1	EFEMP1	0.01	−1.84 (−2.33; −1.35)
P55290	Cadherin-13	CDH13	0.02	−1.75 (−2.11; −1.40)
P98160	Basement membrane-specific heparan sulfate proteoglycan core protein	HSPG2	0.03	−1.63 (−2.07; −1.19)
P04746	Pancreatic alpha -amylase	AMY2A	0.03	−1.57 (−1.95; −1.19)
P01876	Immunoglobulin heavy constant alpha 1	IGHA1	0.04	1.55 (1.32; 1.78)
P02760	Protein AMBP	AMBP	0.04	−1.51 (−1.88; −1.13)
P12830	Cadherin-1	CDH1	0.05	−1.48 (−1.90; −1.07)
Q12907	Vesicular integral-membrane protein VIP36	LMAN2	0.05	−1.46 (−2.10; −0.83)
Q9NPP6	Immunoglobulin heavy chain variant	N/A	0.04	1.58 (1.22; 1.93)
P02766	Transthyretin	TTR	0.05	1.42 (0.97; 1.87)

The protein identification and label-free quantification performed by the MaxQuant software revealed fourteen dysregulated proteins (*p*-value ≤ 0.05) between the tested groups. These proteins are shown in this table along with their *p*-value and effect size. The Cohen’s d fof individual proteins is presented together with the lower and upper 95% confidence interval (CI). Abbreviation: Confidence interval (CI).

**Table 3 cancers-14-02001-t003:** List of mutations mapped on some proteins and respective mutant peptides identified in urine from PCa patients.

Uniprot ID	Protein Name	Gene Name	Mutation Description	Mutation Type	Protein Role in PCa or Other Types of Cancer
P02760	Protein AMBP	AMBP	G238S; E192G; V69M; A286G; P197S; R185Q; G338S; G341A; I198T; V313I; G186R; R185Q	missense	AMBP is an inflammation-regulating protein, associated with human cancers [40,41], including PCa [42,43]. Increased urinary levels [6,42,44,45] but diminished levels in tumor prostate tissue have been reported in PCa patients [46,47,48].
P12830	Cadherin-1	CDH1	H233R; A408E	missense	CDH1 is a protein implicated in cell adhesion, migration, and epithelial-mesenchymal transition [49,50] and its downregulation is correlated with a poor prognosis in PCa patients [51].
Q12805	EGF-containing fibulin-like extracellular matrix protein 1	EFEMP1	V463M	missense	EFMP1 plays a role in cell adhesion and migration, acting as a tumor suppressor in PCa. Diminished EFEMP1 mRNA and protein levels [52] and EFEMP1 promoter hypermethylation were observed in PCa patients [53,54].
P98160	Basement membrane-specific heparan sulfate proteoglycan core protein	HSPG2	V4332I; A1503V; S970F; M638V; Q1062H	missense	HSPG2, found predominantly in the ECM and bone marrow, modulates tumor angiogenesis, proliferation, and differentiation. It is overexpressed in PCa tissues compared to non-malignant tissues, correlating with high GS and PCa cell proliferation and viability [55,56,57].
Q14624	Inter-alpha-trypsin inhibitor heavy chain H4	ITIH4	R866C; G893S	missense	ITIH4 is an acute-phase response protein whose function remains unclear [58]. Research points to a tumor suppressor activity of ITIH4 in human cancers and dysregulation in PCa [43,59].
P07288	Prostate-specific antigen (PSA)	KLK3	C209Y; V55M; G156V; AVCG (47–50); S117P; G87R; L124F; A154T; I179T	Missense; inframe_insertion	PSA is widely used as serum biomarker for PCa. It was approved by the US Food and Drug Administration (FDA) in 1994 [60].
Q12907	Vesicular integral-membrane protein VIP36	LMAN2	G250S; D229N	missense	LMAN2 protein is involved in endoplasmic reticulum to Golgi trafficking of some glycoproteins [61]. Dysregulation of the LMAN2 gene has been indicated in some cancers [62,63,64], while the role in PCa remains obscure. However, raised LMAN2 urinary levels were detected in PCa patients [44].
P41222	Prostaglandin-H2 D-isomerase	PTGDS	L130M	missense	PTGDS is involved in prostaglandins metabolism and lipid transport. The PTGDS gene is downregulated in malignant prostate tissues compared to non-malignant tissues and integrates a signature that predicts relapse after prostatectomy. In vitro, its overexpression increased death and suppressed the growth of PCa cells [65,66].
Q13510	Acid ceramidase	ASAH1	V246A	missense	ASAH1 hydrolyzes ceramide to sphingosine and fatty acid [67] and its protein levels are elevated in tumor prostate tissue [68]. Its increased levels have been suggested as a therapeutic target in PCa as they have been correlated with metastasis establishment and resistance to chemotherapy [69,70].
P08294	Extracellular superoxide dismutase [Cu-Zn]	SOD3	A58T	missense	SOD3 is a known tumor suppressor gene in PCa. It is an antioxidant enzyme that catalyzes the dismutation of the superoxide radical anion [71]. SOD3-reduced levels were reported in PCa patients, and its overexpression in PCa cells prevented cell proliferation, migration, and invasion, suggesting a role as a therapeutic target and predictive marker [72,73].
P09211	Glutathione S-transferase P	GSTP1	I105V	missense	GSTP1 is a known tumor suppressor gene in PCa and is responsible for cellular detoxification through glutathione conjugation [74]. PCa is characterized by loss of GSTP1 function, mostly due to hypermethylation of its regulatory CpG island [75], and it is purported to occur early in prostatic carcinogenesis [76,77].
P10451	Osteopontin	SPP1	A22G	missense	SPP1 is a bone matrix protein involved in bone remodeling, modulation of inflammation, cell adhesion, and migration and angiogenesis [78]. In PCa, SPP1 is associated with metastasis and proliferation [79], lower overall survival and biochemical relapse-free survival, and high GS [80]. Higher SPP1 levels were reported in PCa patients [80,81,82].
P15309	Prostatic acid phosphatase	PAP	G68D	missense	PAP is one of the main secreted proteins by the prostate cells and was the first serum screening marker for PCa. PAP was latter replaced by PSA [83,84].
P25311	Zinc-alpha-2-glycoprotein	ZAG	P187L; A46T	missense	ZAG promotes adipocyte lipolysis, resulting in cancer cachexia [85]. Elevated levels of this protein have been proposed as a serum marker for PCa [86,87], and a significant predictive ability was found for urinary ZAG [8].
Q4ZJI4	Sodium/hydrogen exchanger 9B1	SLC9B1	N70S	missense	SLC9B1 is a Na^+^/H^+^ transporter responsible for preserving cellular homeostasis [88], but this transporter has not yet been correlated with any type of cancer.
Q9P2J8	Zinc finger protein 624	ZNF624	S207F	missense	ZNF624 has not been well studied yet, but in breast cancer was one of the target genes of a microRNA found to be significantly and independently correlated with patient prognosis [89].
Q6EMK4	Vasorin	VASN	R161Q	missense	VASN, an inhibitor of TGF-beta signaling, is upregulated in PCa tissues and stimulates PCa proliferation [90].
P08174	Complement decay-accelerating factor	CD55	S162L	missense	CD55 inhibits the complement system [91]. In PCa, CD55 mediates tumor cells survival and growth [92].

This table shows the UniProt IDs, protein and gene names, mutation site/description and type, and the role of proteins in PCa.

**Table 4 cancers-14-02001-t004:** Results of Polyphen-2 score and prediction for the mapped mutations.

Gene Name	Mutation	Prediction	Score	Sensitivity	Specificity
AMBP	G238S	Probably damaging	1.000	0.00	1.00
AMBP	E192G	Probably damaging	0.75	0.981	0.96
AMBP	V69M	Possibly damaging	0.758	0.85	0.92
AMBP	A286G	Probably damaging	1.000	0.00	1.00
AMBP	P197S	Benign	0.051	0.94	0.83
AMBP	G338S	Probably damaging	0.994	0.69	0.97
AMBP	G341A	Probably damaging	0.958	0.78	0.95
AMBP	V313I	Benign	0.025	0.95	0.81
AMBP	G186R	Probably damaging	1.000	0.00	1.00
AMBP	R185Q	Probably damaging	0.992	0.70	0.97
CDH1	H233R	Possibly damaging	0.831	0.84	0.93
CDH1	A408E	Possibly damaging	0.798	0.84	0.93
EFEMP1	V463M	Probably damaging	0.999	0.14	0.99
HSPG2	V4332I	Benign	0.001	0.99	0.15
HSPG2	A1503V	Probably damaging	1.00	0.00	1.00
HSPG2	S970F	Possibly damaging	0.498	0.88	0.90
HSPG2	M638V	Benign	0.00	1.00	0.00
HSPG2	Q1062H	Benign	0.00	1.00	0.00
ITIH4	R866C	Probably damaging	1	0.00	1.00
ITIH4	G893S	Benign	0.00	1.00	0.00
KLK3	C209Y	Probably damaging	1.000	0.00	1.00
KLK3	G156V	Probably damaging	1.000	0.00	1.00
KLK3	V55M	Probably damaging	0.972	0.77	0.96
KLK3	S117P	Possibly damaging	0.621	0.87	0.91
KLK3	G87R	Benign	0.128	0.93	0.86
KLK3	L124F	Probably damaging	1.000	0.00	1.00
KLK3	A154T	Possibly damaging	0.657	0.86	0.91
KLK3	I 179T	Possibly damaging	0.800	0.84	0.93
LMAN2	G250S	Probably damaging	1.00	0.00	1.00
LMAN2	D229N	Probably damaging	0.983	0.74	0.96
PTGDS	L130M	Probably damaging	1.00	0.00	1.00
ASAH1	V246A	Benign	0.00	1.00	0.00
SOD3	A58T	Benign	0.188	0.92	0.87
GSTP1	I105V	Benign	0.00	1.00	0.00
SPP1	A22G	Possibly damaging	0.611	0.87	0.91
ACP3	G68D	Probably damaging	1.00	0.00	1.00
AZGP1	P187L	Probably damaging	0.94	0.69	0.97
AZGP1	A46T	Benign	0.002	0.99	0.30
SLC9B1	N70S	Benign	0.036	0.94	0.82
ZNF624	S207F	Benign	0.214	0.92	0.88
VASN	R161Q	Benign	0.019	0.95	0.80
CD55	S162L	Probably damaging	0.990	0.72	0.97

## Data Availability

Data generated during Mass Spectrometry analysis is available in the ProteomeXchange Consortium via the PRIDE partner repository with the data set identifier PXD017902.

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
