# Peer review of "Application of Proteogenomics to Urine Analysis towards the Identification of Novel Biomarkers of Prostate Cancer: An Exploratory Study"

_cancers, 2022, doi:10.3390/cancers14082001_

Round 1

Reviewer 1 Report

Thank you very much for giving me this opportunity to review this article.   The authors analyzed protein expression in urine samples from prostate cancer and non-cancer patients. Significant expression changes were detected in several proteins in the discovery cohort, but no significant changes were seen in the testing cohort. On the other hand, mutations were observed in several proteins depending on the presence or absence of prostate cancer.   The followings are comments.   (Major) None.   (Miner) It seems that there are too many tables. Please describe more concisely.

Reviewer 2 Report

Cancers (MDPI)

Application of proteogenomics to urine analysis towards the 2 identification of novel biomarkers of prostate cancer: an exploratory study

Revision:

The authors analyze the urine proteome of PCa patients with a non-invasively approach by a proteogenomic methods to evaluated protein markers for PCa.

The authors use different techniques as mass spectrometry (data were evaluated using two different software), immunoblot and immunoassay.

They also compare the the levels of native and mutant forms of proteins in the urine from PCa patients and try to determine the potential impact of point mutations on function of each protein (PolyPhen-2 tool).

This study could be considered a "pilot study" because of limited number of cases, but the paper is interesting, and the experimental methods are clear (some minor text revisions are necessary).

TEXT Revision:

  • Page4 line 126
    Please replace the “…at 4,000 rpm…”. It is better to express this information in “x g”.
  • Page4 line 145
    Please Please replace “ACN” with “ACN (acetonitrile).
  • Page6 line 226
    Please replace “Ph” with “pH”
  • Page14 line 423
    Please replace “PCA” with “PCA (Principal Component Analysis”
